# The Dietary Intake of Polyphenols Is Associated with a Lower Risk of Severe Lumbar Spinal Stenosis: A Case-Control Analysis from the PREFACE Study

**DOI:** 10.3390/nu14245229

**Published:** 2022-12-08

**Authors:** Emilia Ruggiero, Simona Esposito, Simona Costanzo, Augusto Di Castelnuovo, Marianna Storto, Ettore Carpineta, Chiara Cerletti, Maria Benedetta Donati, Sergio Paolini, Vincenzo Esposito, Giovanni de Gaetano, Gualtiero Innocenzi, Licia Iacoviello, Marialaura Bonaccio

**Affiliations:** 1Department of Epidemiology and Prevention, IRCCS Neuromed, 86077 Pozzilli, Italy; 2Mediterranea Cardiocentro, 80122 Napoli, Italy; 3Department of Analysis Lab Diagnostics, IRCCS Neuromed, 86077 Pozzilli, Italy; 4Neurosurgery Department, IRCCS Neuromed, 86077 Pozzilli, Italy; 5Research Center in Epidemiology and Preventive Medicine (EPIMED), Department of Medicine and Surgery, University of Insubria, 21100 Varese, Italy

**Keywords:** lumbar spinal stenosis, polyphenols, case-control

## Abstract

Polyphenols are naturally occurring compounds endowed with antioxidant and anti-inflammatory properties. We sought to examine the association of dietary polyphenols with the risk of severe lumbar spinal stenosis (LSS), a condition possibly characterized by a high inflammatory component. A case-control study included 156 patients with LSS and indication to surgery and 312 controls, matched (1:2) for sex, age (±6 months), and physical activity. The polyphenol intake was calculated by matching food consumption data from a 188-item food frequency questionnaire with the Phenol-Explorer database regarding the polyphenol content of each reported food. In a multivariable-adjusted logistic regression analysis including lifestyles, sociodemographic factors, and the Mediterranean Diet Score, a 1-standard deviation (SD) increase in dietary polyphenols intake was associated with lower odds of LSS (Odds ratio [OR] = 0.65; 95% CI: 0.47–0.89). Analyses of different polyphenol classes showed that a per 1-SD in the consumption of flavonoids and stilbenes was related to lower LSS risk (OR = 0.57; 95% CI: 0.42–0.78; OR = 0.40; 95% CI: 0.27–0.61, respectively). Further adjustment for the total dietary antioxidant capacity did not modify the strength of these associations. A diet rich in polyphenols is independently associated with a lower risk of severe LSS, possibly through mechanisms that include the anti-inflammatory potential of these bioactive compounds.

## 1. Introduction

Lumbar spinal stenosis (LSS) is a condition characterized by the narrowing of the spinal canal due to age-related changes in facet joints, discs, and ligamentum flavum [1]. Consequently, LSS is more prevalent among middle-aged and elderly individuals, and its main clinical symptoms are low back pain, intermittent claudication, numbness and lower limb pain, and deterioration in the muscle power of the lower limbs as well as standing and walking abilities [2].

The prevalence of LSS in the general population is 11%, while it ranges from 25 to 29% in patients from primary and secondary care, respectively [2], and the annual incidence of adult LSS is estimated at 266 million individuals (5668 and 4501 per 100,000 people in Europe and North America, respectively) [3].

The most common disability of people with LSS is represented by musculoskeletal disorders, which have a strong influence on the quality of life of patients [4], especially on physical status [5,6]; therefore, these patients avoid walking and exhibit sedentary behaviors.

A growing body of literature supports the link between sedentary behaviors and accompanying weight gain, which in turn increases the risk of major chronic diseases, such as obesity, metabolic syndrome, cardiovascular disease, type 2 diabetes, and certain cancers [7,8]. Previous evidence from a case-cohort analysis within the PREdictive FACtors of risk and surgical outcomEs in lumbar spinal stenosis (PREFACE) Study indicated that established cardiovascular disease risk factors, and in particular metabolic factors, may have a role in the pathogenesis of LSS [9], suggesting that inflammation participates in the pathogenesis of this disease.

A healthy and balanced diet is one of the most powerful modifiable risk factors to reduce the inflammation-related burden, which possibly results in reduced long-term incidence of inflammatory-related diseases such as cardiovascular disease, type 2 diabetes, and certain cancers, as well documented in numerous epidemiological studies [10,11,12,13].

In the PREFACE Study, the previous findings had already suggested that a diet rich in fruits and cereals was associated with lower risk of LSS, while a large dietary share of ultra-processed foods, that is, food products largely manufactured through industrial techniques, was directly related to an increased risk of this disease [14]. Cereals and fruits are major sources of polyphenols, naturally occurring bioactive compounds that are involved in the defense against ultraviolet radiation or aggression by pathogens [15]. Research from prospective studies and experimental evidence suggest that polyphenols are highly beneficial for human health [16,17,18,19,20] through mechanisms that include antioxidant, anti-inflammatory, immunomodulatory, anti-tumoral, anti-diabetic, and anti-obesity activities [21].

However, to date, no studies have addressed whether diets rich in polyphenols may be useful for preventing LSS. Therefore, this study sought to test the hypothesis that a diet rich in polyphenols is favorably associated with the risk of severe LSS; to this end, a case-control analysis was performed using data from the cohort of patients of the PREFACE Study [9,14].

## 2. Methods

### 2.1. Case Patients

A total of 264 consecutive patients, aged ≥35 years, with symptomatic LSS, confirmed by imaging, and eligible for surgery at the Neurosurgery Department of the IRCCS (Istituto di Ricovero e Cura a Carattere Scientifico) Neuromed in Italy, were enrolled in the PREFACE Study from 2016 to 2018 [9,14].

LSS is defined by the presence of intermittent neurogenic claudication or signs of chronic neurogenic compression associated with the presence of a central or lateral compression of the cauda equina upon imaging, reviewed by neurosurgeons at the IRCCS Neuromed.

Inclusion criteria were age ≥18 years, a formal diagnosis of lumbar spinal stenosis based on neuroimaging scan, being considered not to have improved with conservative treatments for almost six months, classified as grade C or D on the Schizas’s scale [22], and being scheduled for surgery at the Neurosurgery Department of the IRCCS Neuromed.

Exclusion criteria were associated clinical conditions responsible for functional disability (cervical myelopathy, peripheral neuropathy, inflammatory rheumatoid disease), a previous history of lumbar canal decompression, spine instability, pregnancy, and refusal to sign the informed consent at the time of the investigation.

All patients provided written informed consent to be enrolled in the study.

Anthropometric measurements and administration of questionnaires (anamnestic and dietary) were completed before surgery. Trained research personnel at the Department of Epidemiology and Prevention at the IRCCS Neuromed took anthropometric measurements using methods that had been standardized beforehand during preliminary training sessions.

Trained interviewers administered the questionnaires to collect personal and clinical information including socio-economic status, physical activity, physio/pathological medical history risk factors, and drug use.

The recruitment was organized between 8.00 and 11.00 a.m. in the Neuromed clinic.

For the purpose of the present analyses, we excluded LSS cases with incomplete/missing data related to diet (*n* = 90) or reporting implausible energy intakes (<800 kcal/d in men and <500 kcal/d in women or >4000 kcal/d in men and >3500 kcal/d in women; *n* = 4). The final sample consisted of 156 LSS cases. The study was approved by the Ethics Committee of the IRCCS Neuromed (Pozzilli, Italy; protocol number: 05202016).

### 2.2. Control Subjects

The matched-control participants included 312 subjects without any reported spine degenerative disease or clinical sign of back disease identified among the participants of the Moli-sani Study, a large population-based cohort of men and women (aged ≥35 years) randomly enrolled from city hall registries by multistage sampling, from the general population of the Molise region, where IRCCS Neuromed is located, between 2005–2010 [23]. For each LSS case, we matched two controls by sex, age (±6 months), and physical activity (sedentary, low, physically active).

### 2.3. Assessment of Covariates

Evaluation of the covariates was conducted with similar procedures, both in cases and controls [9,14], with the exception of physical activity. Amongst participants with LSS, habitual leisure-time physical activity (PA) was ascertained through the question “How was your leisure-time physical activity in the last six months?” with three possible answers (i.e., sedentary, moderate, or physically active). Leisure-time physical activity in the Moli-sani Study was assessed by an interviewer-administered structured questionnaire [23]; the total time (minutes per week) spent in sport, walking, and gardening was categorized into the following categories: (a) sedentary (e.g., none of the above activities); (b) moderate (i.e., at least 4 h/week in walking or gardening); physically active (e.g., any type of sport).

Occupation was categorized as non-manual, manual, retired non-manual, retired manual, and other/unclassified. Subjects were classified as never, current, or former smokers (reported not having smoked at all over the previous 12 months or more). Height and weight were measured on a standard beam balance scale with an attached ruler, in subjects wearing no shoes and only light indoor clothing, and body mass index (BMI) was calculated as kg/m^2^ and then grouped into three categories as normal (<25 kg/m^2^), overweight (≥25 < 30 kg/m^2^), or obese (≥30 kg/m^2^) [24].

Participants were considered to have hypertension, hyperlipidemia, or diabetes if they reported having been treated with disease-specific drugs. The participants’ history of cardiovascular (angina, stroke and myocardial infarction), peripheral artery disease, and cancer included self-reported diagnosis.

### 2.4. Dietary Assessment

Both for LSS cases and control participants, data on food intake during the year before enrolment were collected through the validated Italian version of the EPIC food frequency questionnaire (FFQ) [25]; the same FFQ was interviewer-administered to Moli-sani participants (controls) and self-administered in the group of LSS patients. The Italian version of the EPIC FFQ contains 14 sections (i.e., pasta/rice, soup, meat, excluding salami and other cured meats, fish, raw vegetables, cooked vegetables, eggs, sandwiches, salami and other cured meats, cheese, fruit, bread/wine, milk/coffee/cakes, and herbs/spices) with 248 questions concerning 188 different food items. The frequencies and quantities of each food were then linked, using a specifically designed software [26], to the Italian Food Tables [27] to obtain estimates of daily intakes of macro- and micronutrients plus energy.

A total of 188 food items included in the EPIC-FFQ were studied as potential food sources of polyphenols. The consumption of total polyphenols and their classes (phenolic acids, flavonoids, stilbenes, lignans, and other polyphenols) was considered.

The Phenol-Explorer database was used to estimate the total intake of polyphenols and their classes and subclasses [28] according to the following: (1) a total of 188 food items from the FFQ were evaluated as potential food sources of polyphenols; (2) when an item from the FFQ included several foods (e.g., oranges and grapefruit), the proportion of intake was calculated; (3) the polyphenol content (of each of five classes) in 100 g of each food item was obtained from the Phenol-Explorer database; and (4) finally, the individual subclasses of polyphenol intake from each food were calculated by multiplying the content of each polyphenol by the daily consumption of each food.

Using this information and the daily consumption of each food source (g), the total intakes of five classes of polyphenols were calculated as follows: flavonoids (mg/d), flavones (mg/d), lignans (mg/d), stilbenes (mg/d), phenolic acids (mg/d), and other polyphenols (mg/d). The total polyphenol intake was calculated as the sum of all individual polyphenol intakes from the food sources used in the FFQ. The same procedure was used for dietary data in both cases and controls.

The total antioxidant capacity (TAC) of the diet was measured in foods via the use of three different assays: the trolox equivalent antioxidant capacity (TEAC, mmol Trolox) assay, measuring the antioxidants’ ability to reduce a radical cation in both lipophilic and hydrophilic conditions [29]; the radical-trapping antioxidant parameter (TRAP, mmol Trolox) and ferric-reducing antioxidant power (FRAP, mmol Fe^2+^) assays, evaluating the chain-breaking antioxidant potential [30] and the reducing power of the sample, respectively [31]; and, finally, adherence to the traditional Mediterranean diet (MD) was defined according to the Mediterranean Diet Score (MDS) developed by Trichopoulou et al. [31] by assigning one point to healthy foods’ (such as fruits and nuts, vegetables, legumes, fish, and cereals) monounsaturated (MUFA) to saturated fat (SFA) ratio, whose consumption was above the sex-specific medians of intake of the Moli-sani Study population; foods presumed to be detrimental (meat and dairy products) were scored positively if their consumption was below the median. All other intakes received zero points. For ethanol, men who consumed 10–50 g/d and women who consumed 5–25 g/d received one point; otherwise, the score was zero. The MDS ranged from 0 to 9 (the latter reflecting maximal adherence) [32].

### 2.5. Statistical Analysis

With the available sample size of 156 case patients and 312 control individuals, we were able to detect true odds ratios for diseases lower than 0.52 or greater than 1.91 in exposed subjects (individuals with polyphenols intake over the population median) relative to unexposed subjects with a probability (power) of 80%, with a Type I error probability 0.05.

The main characteristics of cases and controls were reported as percentages for categorical variables or mean values and standard deviations (±SD) for continuous traits. The conditional-to-match differences between cases and controls were evaluated by general linear models (PROC GENMOD and PROC GLM in SAS for categorical and continuous variables, respectively).

The distribution of missing values was as follows: educational level (*n* = 1), BMI (*n* = 16), abdominal obesity (*n* = 16), smoking habit (*n* = 1), history of cardiovascular disease (*n* = 1), diabetes (*n* = 1), hyperlipidemia (*n* = 1), and hypertension (*n* = 1). We used regression-based imputation to impute missing data, avoiding the bias introduced by not-at-random missing data patterns.

Multivariable conditional logistic regression analysis was used to quantify the association of dietary factors with LSS risk. Odds ratios (ORs) and their 95% confidence intervals (CIs) were calculated using conditional-to-match (for age, sex, and physical activity) logistic regression. Potential confounders were defined a priori and identified based on existing literature, rather than deferring to statistical criteria [33], on the basis of their previously documented associations both with diet and LSS.

Socioeconomic status is reportedly associated both with diet quality [34] and LSS [35]; smoking status, leisure-time physical activity, and BMI have an impact on chronic low-grade inflammation [36], which is critical to LSS, and they tend to cluster with other behaviors such as dietary habits [37]; finally, main chronic diseases and other health conditions possibly result from a prolonged pro-inflammatory state, which may have been promoted by unhealthy health-related behaviors, including diet [36].

Three main multivariable models were fitted: (a) model 1 included age, sex, physical activity (matched variables), and energy intake; (b) model 2 was additionally adjusted for socioeconomic factors (occupation, educational level, place of residence), lifestyle (smoking status and BMI), and chronic disease and health conditions (cancer, hypertension, diabetes, hyperlipidemia, and peripheral artery disease); (c) model 3 was the same as model 2, additionally including the MDS. This multivariable-adjusted model 3 served as the reference to test whether the association of dietary polyphenols with the risk of LSS could be explained by the total antioxidant capacity of the diet as reflected by three different assays (i.e., TEAC, TRAP, and FRAP) that were alternately included in the multivariable-adjusted model 3. Data analyses were generated using SAS/STAT software, version 9.4 (SAS Institute Inc., Cary, NC, USA).

## 3. Results

The main characteristics of cases and matched-controls are shown in Table 1. Cases were more likely to be manual workers and more obese and tended to have a higher prevalence of diabetes, hypertension, and hyperlipidemia.

The average intake of polyphenols in the diet was 595.3 mg/d (±274.1) in cases and 677.8 mg/d (±218.0) in controls (*p* value = 0.0002) (Table 2). Among polyphenol classes, the LSS cases had lower intakes of flavonoids, stilbenes, and other polyphenols, as compared to matched controls (Table 2).

The top foods contributing to the intake of polyphenol classes of cases and matched-controls are reported in Table 3, with no substantial differences between cases and controls. In univariate analysis (matched for age, sex, and physical activity), a 1-SD increment in the total polyphenol eaten (281.0 mg/d) was associated with a decreased risk of LSS (OR = 0.63; 95% CI 0.49–0.82; Table 4, Model 1), and the results remained substantially unchanged in multivariable analyses (OR = 0.66; 95% CI 0.48–0.90; Table 4, Model 2); further adjustments for the MDS did not alter the strength of the association (OR = 0.65; 95% CI 0.47–0.89; Table 4, Model 3). Per 1-SD increment of total flavonoid (173.8 mg/d), the stilbenes (9.1 mg/d) and other polyphenols (23.9 mg/d) intakes were inversely associated with the risk of LSS (OR = 0.57, 95% CI 0.42–0.78; OR = 0.40, 95% CI 0.27–0.61; and OR = 0.30, 95% CI 0.20–0.47, respectively; Table 4, Model 3). The main results were confirmed when the dietary exposures were modelled as thirds of the intake (Figure 1). The alternate inclusion of different assays reflecting the antioxidant capacity of the diet did not modify the magnitude of the association between the total dietary polyphenol content and the risk of LSS (Table 4).

## 4. Discussion

The main aim of this case-control study was to test the hypothesis that a diet rich in polyphenols could be favorably associated with the risk of severe LSS, a condition possibly characterized by an altered inflammatory status. Our findings were that high amounts of polyphenols in the diet are linked to lower odds of LSS, independent of the overall diet quality. Some specific classes of polyphenols, such as flavonoids and stilbenes, which are abundant in red wine, seasonal fruit, and tea, were the main drivers of this favorable association.

LSS is a debilitating and degenerative condition which typically affects adults at older age, and it also has a considerable impact on the risk for chronic diseases of inactivity [38]. More recently, LSS was reported to be associated with a 1.6-fold increased risk of severe disability and mortality and a 1.5-fold increased risk of mortality in community-dwelling older adults [39]. This condition is projected to increase with the aging of the population and to be more frequently diagnosed with the increased use of advanced imaging [40].

Multiple factors can contribute to the development of LSS, and these can act synergistically to exacerbate the condition [41].

Recently, findings from the PREFACE Study suggested that the most common risk factors for cardiovascular disease, and, in particular, metabolic risk factors were associated with increased risk of LSS [9].

Previous evidence also provided support for a link between obesity-related systemic inflammation and musculoskeletal pain [42,43,44,45]. Consistently, a pilot study demonstrated that a lifestyle intervention, such as pedometer-based physical activity and nutritional education, has the potential to provide a non-surgical management option for people with LSS [38,46].

Likewise, it is well-known that a nutritionally adequate diet is a key determinant of human health, which is effective in reducing the risk of a variety of non-communicable diseases [47,48], through several mechanisms that include the favorable modulation of the inflammatory state [49,50].

Previous analyses from the PREFACE Study showed that a higher consumption of foods of a traditional MD, such as fruits, nuts, and cereals, was associated with lower odds of severe LSS; on the contrary, a large dietary share of ultra-processed food (i.e., food products that have been manufactured through industrial techniques) was found to be related to higher odds of developing severe LSS [14]. A traditional MD, as well as many other healthy plant-based diets, is rich in foods that are a major source of polyphenols; the latter are secondary metabolites found mainly in various fruits, vegetables, cereals, and beverages such as cocoa, red wine, and tea [15].

Polyphenols were found to act on both the inflammatory processes and endothelial dysfunction, and their beneficial effects are also attributed to their antioxidant properties and free-radical-scavenging activities [49] and to their capacity to donate hydrogen atoms or electrons or to chelate metal cations [50].

To see whether the health advantages of polyphenols against LSS risk could be explained by their antioxidant potential, we adjusted the main analyses for the total antioxidant capacity of the diet, but the results remain substantially unchanged. These findings suggest that the protection of polyphenols against severe LSS is unlikely to be explained by their antioxidant activity, and that the explanation should be sought instead in their anti-inflammatory action, which is largely documented [51,52], also in epidemiological settings [53]. Prior analyses from the Moli-sani Study showed that the association of dietary polyphenols with biological aging, a novel predictor of cardiovascular disease risk, was attributable to mechanisms that go beyond the antioxidant activity of these compounds [54]; similarly, in a cohort of postmenopausal Polish women, the dietary polyphenol intake, but not the total dietary antioxidant capacity, was inversely related to cardiovascular disease [55].

In the present study, the favorable relationship of dietary polyphenols to the risk of severe LSS emerged with some specific classes, such as total flavonoids and stilbenes, while it was not seen for total phenolic acids and lignans. These findings may be ascribed to the differences in the structure of each class, which depends on the numbers and position of hydroxyl groups in the molecule [56].

### 4.1. Strengths

To our knowledge, this study is the first to evaluate the association between dietary polyphenols and the risk of severe LSS. This study has several strengths, including the use of a comprehensive and reproducible FFQ [25] and the use of food composition databases to estimate polyphenol intake. Matching for age, sex, and physical activity was intended to eliminate confounding; however, the main potential benefit of matching in this case-control study was a gain in efficiency. Moreover, our analyses were controlled for a large number of covariates that possibly minimize confounding.

### 4.2. Limitations

Several limitations are also inherent to this study. As is common to all observational studies, there may have been a recall bias associated with the reporting of dietary intake data and other factors. Moreover, the variety of polyphenols composition is complex, and the use of FFQs may lead to measurement errors; although we have used one of the most comprehensive food composition databases on polyphenols to date [30], measurement error in collecting and estimating dietary polyphenol intake remains an issue. In addition, the method of the administration of FFQs and questionnaires to evaluate physical activity levels differed between cases and controls, which could also be a source of bias. However, there is evidence supporting the agreement between self-administered questionnaires and dietetic interviews [57,58]. Another limitation is that this analysis was restricted to severe LSS, and this limits the generalizability of results related to LSS not requiring surgery.

Finally, a common limitation of population-based case-control studies that include a non-screened population as a control is that investigators may be unaware of whether some undiagnosed LSS occurred in the control group; however, Moli-sani participants reporting diagnostic imaging studies for back pain were excluded, and the rate of potentially undiagnosed LSS would be low.

## 5. Conclusions

The results from this case-control study suggest that a diet rich in polyphenols is associated with a lower risk of severe LSS. In particular, a favorable relationship emerged with some specific polyphenol classes, such as flavonoids and stilbenes. Although coming from a case-control analysis, which has inherent limits, these findings possibly contribute to shedding light on potential diet-related mechanisms underlying the onset of severe LSS, and therefore represent a good basis for the more effective prevention, and eventually treatment, of this disease.

Studies examining the urinary levels of polyphenols and the risk of LSS may be helpful in exploring this association further, as will studies with larger sample sizes and a prospective design.

## Figures and Tables

**Figure 1 nutrients-14-05229-f001:**
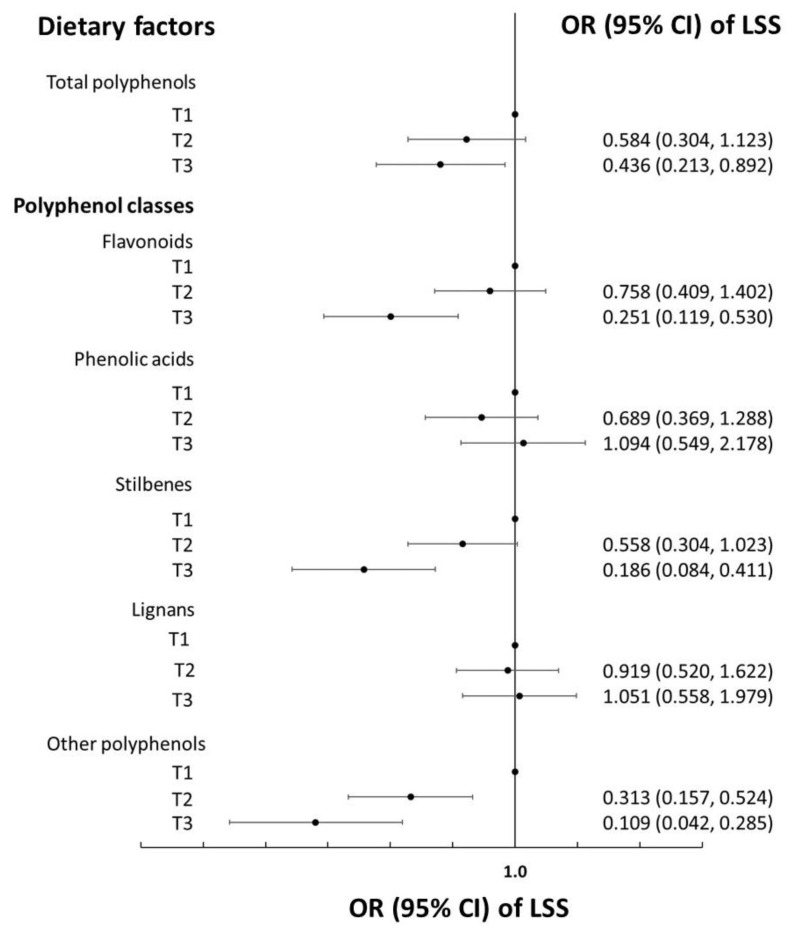
Dietary polyphenols (tertiles of) and risk of lumbar spinal stenosis (LSS) in the PREFACE case-control study. Odds ratios (ORs) and their 95% confidence intervals (CIs) were calculated using conditional-to-match (for age, sex, and physical activity) logistic regression adjusted for energy intake, occupation, educational level, smoking habit, place of residence, body mass index, cancer, hypertension, diabetes, hyperlipidemia, peripheral artery disease, cardiovascular disease, and the Mediterranean Diet Score.

**Table 1 nutrients-14-05229-t001:** Main characteristics of Lumbar Spinal Stenosis (LSS) cases and matched controls analyzed in the PREFACE Study.

	LSS Cases *n* = 156	Controls *n* = 312	*p*-Value
Sex (Men)	69.2	69.2	Matching variable
Age (years; means, SD)	66.2 (9.2)	66.2 (9.2)	Matching variable
Physical activity level (%)			Matching variable
Sedentary lifestyle	67.3	67.3	
Low active lifestyle	27.6	32.1	
Physically active lifestyle	5.1	0.6	
Educational level (%)			0.87
Up to lower secondary	62.2	64.4	
Upper secondary	29.5	27.9	
Post-secondary	8.3	7.7	
Occupation (%)			<0.001
Non-manual	12.2	12.2	
Manual	14.7	6.7	
Retired (Non-manual)	21.2	17.9	
Retired (Manual)	34.6	22.8	
Other/Unclassified	17.3	40.4	
Place of residence (%)			0.92
Rural	36.5	38.8	
Urban	63.5	61.2	
Smoking habits (%)			0.13
Never	37.8	41.3	
Current	26.9	18.6	
Former	35.3	40.1	
Body mass index (%)			<0.001
Normal	17.3	18.6	
Overweight	40.4	44.9	
Obese	37.2	36.5	
Cardiovascular disease (%)	14.7	13.1	0.63
Cancer (%)	10.3	5.1	0.069
Diabetes (%)	22.4	10.3	<0.001
Hypertension (%)	69.2	44.9	<0.001
Hyperlipidemia (%)	26.9	15.7	0.0034
Peripheral artery disease (%)	2.6	1.0	0.18

SD = standard deviation. Univariable *p*-values (matched for age, sex, and physical activity).

**Table 2 nutrients-14-05229-t002:** Dietary factors in Lumbar Spinal Stenosis (LSS) cases and matched controls.

Dietary Factors	LSS Cases *n* = 156	Controls *n* = 312	*p*-Value
	Means (SD)	Means (SD)	
Total polyphenols intake (mg/d)	595.3 (274.1)	677.8 (218.0)	0.0002
Polyphenol classes (mg/d)			
Flavonoids	216.0 (160.4)	282.2 (173.8)	<0.0001
Lignans	23.7 (14.5)	25.6 (12.9)	0.13
Stilbenes	3.3 (6.3)	7.6 (9.1)	<0.0001
Phenolic acids	303.5 (128.7)	301.4 (121.9)	0.84
Other polyphenols	48.8 (25.7)	61.0 (23.9)	<0.0001
Total antioxidant capacity			
TEAC	5.8 (2.5)	6.3 (2.8)	0.021
TRAP	8.5 (3.8)	9.1 (4.1)	0.064
FRAP	17.5 (7.3)	18.7 (8.2)	0.061
Mediterranean Diet Score	4.0 (1.8)	4.1(4.5)	0.84

Means and *p* values are adjusted for energy intake (kcal/d) and for matching design. TEAC = Trolox equivalent antioxidant capacity. TRAP = Radical-trapping antioxidant parameter assay. FRAP = Ferric-reducing antioxidant power assay.

**Table 3 nutrients-14-05229-t003:** Main food contributors to the intake of polyphenol classes in LSS cases and controls.

Polyphenol Classes ^1^	LSS Cases (*n* = 156)	Controls (*n* = 312)
Flavonoids	Seasonal fruit (34.9%), red wine (17.3%), and tea (6.9%).	Seasonal fruit (39.9%), red wine (31.3%), and citrus fruit (10.0%).
Lignans	Seasonal fruit (42.0%), citrus fruit (19.6%), and cabbages (12.7%).	Seasonal fruit (41.5%), citrus fruit (21.7%), and cabbages (11.2%).
Stilbenes	Red wine (41.8%), seasonal fruit (35.9%), and white wine (12.9%).	Red wine (63.0%), seasonal fruit (27.0%), and white wine (4.5%).
Phenolic acids	Coffee (51.7%), seasonal fruit (22.8%), and fruit and orange juices (3.2%).	Coffee (44.7%), seasonal fruit (29.9%), and artichoke (3.2%).
Other polyphenols	Bread (36.8%), olive oil (31.7%), and red wine (6.4%).	Bread (51.8%), olive oil (24.5%), and red wine (10.3%).

^1^ % of total intake within each polyphenol class. The first three main food contributors in each polyphenols class are given; a lower number of foods indicates the absence or the marginal contribution of other food contributors that contained polyphenols in the subclass considered.

**Table 4 nutrients-14-05229-t004:** Association of dietary polyphenols with risk of Lumbar Spinal Stenosis.

Dietary Factors	Model 1	Model 2	Model 3	Model 3 + TEAC		Model 3 + TRAP		Model 3 + FRAP	
	OR (95% CI)	OR (95% CI)	OR (95% CI)	OR (95% CI)	Δ (%) *	OR (95% CI)	Δ (%) *	OR (95% CI)	Δ (%) *
**Per 1-SD increment**									
Total polyphenols	0.63 (0.49–0.82)	0.66 (0.48–0.90)	0.65 (0.47–0.89)	0.56 (0.33–0.94)	34.6	0.59 (0.38–0.93)	22.5	0.55 (0.34–0.88)	38.8
Polyphenol classes									
Flavonoids	0.58 (0.45–0.75)	0.59 (0.44–0.80)	0.57 (0.42–0.78)	0.53 (0.35–0.78)	12.9	0.57 (0.40–0.80)	0.0	0.54 (0.37–0.78)	9.6
Phenolic acids	1.02 (0.82–1.28)	1.07 (0.82–1.40)	1.07 (0.81–1.40)	1.57 (1.07–2.31)	-	1.66 (1.10–2.51)	-	1.57 (1.06–2.34)	-
Stilbenes	0.44 (0.31–0.61)	0.41 (0.27–0.61)	0.40 (0.27–0.61)	0.28 (0.16–0.48)	38.9	0.34 (0.21–0.55)	17.7	0.31 (0.19–0.51)	27.8
Lignans	0.86 (0.70–1.06)	0.99 (0.78–1.26)	0.99 (0.77–1.27)	1.07 (0.82–1.39)	-	1.03 (0.80–1.34)	-	1.05 (0.80–1.36)	-
Other polyphenols	0.33 (0.23–0.47)	0.31 (0.20–0.47)	0.30 (0.20–0.47)	0.31 (0.20–0.48)	2.7	0.31 (0.20-0.48)	2.7	0.31 (0.20–0.47)	2.7

Odds ratio (ORs) with 95% confidence interval (95%CI). Model 1 = matched for age, sex, physical activity, and energy intake. Model 2 = matched for age, sex, physical activity, and energy intake and further adjusted for occupation, educational level, place of residence, smoking habit, body mass index, cancer, hypertension, diabetes, hyperlipidemia, peripheral artery disease, and cardiovascular disease. Model 3 = Model 2 plus the Mediterranean Diet Score (range 0–9). TEAC = Trolox equivalent antioxidant capacity. TRAP = Radical-trapping antioxidant parameter assay. FRAP = Ferric reducing antioxidant power assay. * Δ (%), attenuation, representing the proportion of the Polyphenol–LSS association explained by each assay estimating total antioxidant capacity (TAC) of the diet (i.e., TEAC, TRAP or FRAP); and obtained as = 100 × (β Model 1 + TAC assay − β Model 1)/(β Model 1). % attenuation is calculated only for statistically significant associations.

## Data Availability

The data underlying this article will be shared on reasonable request to the corresponding author. The data are stored in an institutional repository (https://repository.neuromed.it (accessed on 1 November 2022)), and access is restricted by the ethical approvals and the legislation of the European Union.

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
