# Peer review of "The Dietary Intake of Polyphenols Is Associated with a Lower Risk of Severe Lumbar Spinal Stenosis: A Case-Control Analysis from the PREFACE Study"

_nutrients, 2022, doi:10.3390/nu14245229_

Round 1
Reviewer 1 Report
This is an interesting paper, and the authors did a good analysis with a robust methodology.
However, I have some concerns:
1) I understand that the LSS patients came from the Neuromed study. However, the authors should specify with precision the inclusion criteria for LSS also in this study. Referring to the previous study is not enough for readers’ clarity.
How the patients were extracted from the Neuromed study?
Were those patients reviewed by someone (i.e., spine surgeon, neuroradiologist)?
What kind of LSS severity score were adopted? Why an LSS classification score was not adopted for the sake of statistical analysis? It would be interesting to run multivariate models with such a stratification.
(please see: Schizas C, Theumann N, Burn A, Tansey R, Wardlaw D, Smith FW, Kulik G. Qualitative grading of severity of lumbar spinal stenosis based on the morphology of the dural sac on magnetic resonance images. Spine (Phila Pa 1976). 2010 Oct 1;35(21):1919-24. doi:0.1097/BRS.0b013e3181d359bd. PMID: 20671589).
I mean, not all the LSS are the same. As a spine surgeon, I consider LSS a pathology of a mixed and multifactorial origin (a coexistence of disc degeneration and circumferential protrusion, degenerative deformity, facets hypertrophy, degenerative listhesis, etc.…). Therefore not all LSS may clearly benefit from polyphenols’ increased intake. In my opinion, it may depend on the kind of stenosis.
As it is written, I feel there is a huge selection bias that compromises the author’s results.
2)How adjusting factors for the multivariate models were chosen? Please support your choice with references.
3)Please correct the BMI part: “Height and weight were measured, and body mass index (BMI) was calculated as kg/m2 and then grouped into 3 categories as normal (<25 kg/m2), overweight (≥25 < 30 kg/m2), or obese (≤30 kg/m2).”
4) the author’s conclusions are too sensationalistic with their results, and because of the consideration made above, I think a more speculative approach may be safer.
Reviewer 2 Report
Dear Authors,
I would like to express my gratitude regarding the opportunity to review this manuscript.
At this stage the manuscript requires considerable improvements. Below suggestions with page indication (no lines in the document):
P1 - Title, Authors and Affiliations – Please reformulated considering the journal template and instructions for authors. The template for example is from 2021.
P1 - Please reformulate ABSTRACT with “abstract” in lowercase, without subtopics.
P2 – “1. Introduction”, not INTRODUCTION. Please carefully review all manuscript considering the journal template and instructions for authors. In this particular, all topics and subtopics.
P2 – Text is not justified, this is mandatory. Please correct.
P2 – Please describe “IRCCS” in full in the first appearance in the text.
P2 – Please clearly describe that all subjects signed an informed consent. It is also important to clearly indicate the inclusion and exclusion criteria.
P2 – “Height and weight were measured, and body mass index (BMI) was calculated as kg/m2 and then grouped into 3 categories as normal (<25 kg/m2), overweight (≥25 < 30 kg/m2), or obese (≤30 kg/m2)” Please provide reference to support the criteria.
P2 – Please indicate the instruments, procedures and conditions of data collection (place, time of day, clothes used by subjects, how supervised – experience and academic background, and other details that should be described).
P3 – “(i.e., pasta/rice, soup, meat (excluding salami and other cured meats)” Please correct.
P4 – Please review the text “ferric reducing-antioxidant power
(FRAP, mmol Fe2+) assays, evaluating the chain-breaking antioxidant potential [29] and the reducing power of the sample, respectively [30].”
P4 – Please correct.
P4 – In statistical analysis, please describe de sample power.
P4 – It is suggested to remove “(Table 1)”, the indication of table is already in the same paragraph.
P5 – Tables 1, 2 and 3 content should consider considering the journal template and instructions for authors. Please correct (namely, but not only, the type and size of letter).
P6 – “595.3”. Some units in the first pages are presented with “,”. It is suggested that units are presents with “.” in all manuscript.
P6 – Text and spaces between tables 2 and 3 should be corrected.
P6 – Table 3 – Please review “.;”.
P7 – Paragraphs are suggested in the text before figure 1. Please note that in this page and afterwards, the headers and footers of the pages require correction.
P7 – Figure 1 content should consider the journal template and instructions for authors. Please correct.
P8 – Table 4 content should consider the journal template and instructions for authors. Please correct.
P9 – The discussion section should start with the aims of the study followed by the main findings, which afterwards should be discussed with literature references.
P9 – “Mediterranean Diet”, already previously abbreviated.
P9 – More than one time in this page and in other pages, more than one space after end point. Please review in all manuscript.
P9 – The discussion section should be more developed, and the inclusion of more references considered.
P10 – “[45, 46].” Please correct.
P10 – Please describe the “Author Contributions:” according to the journal template and instructions for authors.
P10 – Please include “Institutional Review Board Statement”; “Informed Consent Statement” and “Conflicts of Interest” according to the journal template and instructions for authors.
P10-11 – Please format the appendix indication considering the journal template and instructions for authors.
P11 – Please double check all the references considering the journal template and instructions for authors. Some examples: All journals abbreviated in italic, volume number in italic, DOI´s indication and other details.
Please carefully review the document after considering the above indications/suggestions and present the next version with lines indication, aiming a more precise review. Thank you.
Reviewer 3 Report
Congratulations to the authors. The study is very important. I am presenting some comments, as a contribution, to improve the presentation of the manuscript. Please, find in the attached file my suggestions/comments.
